# Cooperative Multi-Objective Optimization of DC Multi-Microgrid Systems in Distribution Networks

**Zhiwen Xu, Changsong Chen \*, Mingyang Dong, Jingyue Zhang, Dongtong Han and Haowen Chen**

State Key Laboratory of Advanced Electromagnetic Engineering and Technology, School of Electrical and Electronic Engineering, Huazhong University of Science and Technology, Wuhan 430074, China; xuzhiwen@hust.edu.cn (Z.X.); dongmingyang@hust.edu.cn (M.D.); m202071581@hust.edu.cn (J.Z.); m202071541@hust.edu.cn (D.H.); m201971407@hust.edu.cn (H.C.)

**\*** Correspondence: ccsfm@hust.edu.cn

**Featured Application: A cooperative multi-objective optimization model of a DC multi-microgrid that considers across-time-and-space energy transmission of EVs is established to improve the economy of the system, decrease the loss of the distribution network, and reduce carbon emissions.**

**Abstract:** By constructing a DC multi-microgrid system (MMGS) including renewable energy sources (RESs) and electric vehicles (EVs) to coordinate with the distribution network, the utilization rate of RESs can be effectively improved and carbon emissions can be reduced. To improve the economy of MMGS and reduce the network loss of the distribution network, a cooperative double-loop optimization strategy is proposed. The inner-loop economic dispatching reduces the daily operating cost of MMGS by optimizing the active power output of RESs, EVs, and DC/AC converters in MMGS. The outer-loop reactive power optimization reduces the network loss of the distribution network by optimizing the reactive power of the bidirectional DC/AC converters. The double-loop, which synergistically optimizes the economic cost and carbon emissions of MMGS, not only improves the economy of MMGS and operational effectiveness of the distribution network but also realizes the low-carbon emissions. The Across-time-and-space energy transmission (ATSET) of the EVs is considered, whose impact on economic dispatching is analyzed. Particle Swarm Optimization (PSO) is applied to iterative solutions. Finally, the rationality and feasibility of the cooperative multi-objective optimization model are proved by a revised IEEE 33-node system.

**Keywords:** DC multi-microgrid system; carbon emissions; economic dispatch; across-time-and-space energy transmission; cooperative multi-objective optimization

## 1. Introduction

Since the national carbon neutrality and carbon peak requirements have been put forward [1], low carbon emissions and new energy have become hot research topics [2,3]. It is a trend to replace petrol vehicles with electric vehicles (EVs) and replace regional large-scale power grids with microgrids (MGs) containing renewable energy sources (RESs) [4,5]. As the output of RESs is intermittent and uncertain, the MGs need to coordinate with the distribution network to centrally regulate the RESs, which is a challenge to the operation mode of the traditional power system. With the popularity of EVs, the burden of the distribution network will greatly be increased. Additionally, the safe operation of the distribution network will be threatened if EVs are charged in the distribution network without control.

The research on the charging and discharging dispatching strategy of EVs is mainly from the view of the economy [6,7]. Many studies have considered charging/discharging strategies of EVs but overlooked the energy storage characteristics of EVs. Through the

bidirectional Vehicle-to-grid (V2G) technology, EVs can also deliver electrical energy to the grid by discharging, and improve the operation of the grid [8–10].

Some research combines EVs with distributed RESs in the MG. In [11], an optimization method for the operation route and charging/discharging time of EVs is proposed, which uses the timely charging/discharging of EVs to consume the output of RESs and reduce the volatility of the equivalent load. In [12], the MG energy management strategy is discussed from the perspective of system operating cost and the consumption efficiency of RESs, and V2G technology has been applied. In [13], the structure and parameter design of the system have been discussed, and the actual MG system using V2G technology has been studied. However, most of the energy scheduling in MGs and the distribution network is to adjust the output of active power.

Some research focuses on the economic dispatch of the multi-microgrid system. In [14], an interconnected multi-microgrids (IMMGs) system using various complementary power sources effectively coordinates the energy sharing/trading among the MGs and the main grid to improve energy efficiency. In [15], A probabilistic modeling of both small-scale energy resources (SSERs) and load demand at each microgrid (MGs) is performed to determine the optimal economic operation of each MG with minimum cost based on the power transaction between the MGs and the main grid. The above does not consider the reactive power exchange between MMG and the main grid.

In the current research on reactive power exchange and network loss, most studies focus on the reactive power of a single distribution network. In [16], the trend of reactive power demand in the distribution network is evaluated. Reactive power demand management plays an important role in the cost-effectiveness and stable operation of the distribution network. A multi-objective planning algorithm for reactive power compensation of radial distribution networks is proposed in [17], which uses unified power quality conditioner (UPQC) compensation for load reactive power to reduce network loss. In [18], the solid-state transformer (SST) is used to supply the load reactive power demand and inject reactive power into the grid, which reduces network losses in a radial distribution network.

Some research focuses on the impact of reactive power optimization on the loss of MG. In [19], a distributed, leaderless and randomized algorithm is proposed, which controls the microgenerators in the island-operated MG system to compensate for reactive power and reduce power distribution loss in MG. A generalized approach for probabilistic optimal reactive power planning is proposed in [20], which can reduce the annual energy losses of the grid-connected MG system.

These papers mentioned above give less consideration to the collaborative optimization of MG clusters and the distribution network. To solve the above problems, a cooperative multi-objective optimization model of a DC multi-microgrid system (MMGS) including RESs, EVs, and DC/AC converters is established. The goal of the model is to obtain the optimal MMGS economic cost and the network loss of the distribution network. The main contributions of this paper are as follows:

1. A grid-connected MMGS containing RESs and EVs is constructed, where RESs, EVs, MGs and distribution networks are combined, bidirectional V2G technology is used and the across-time-and-space energy transmission (ATSET) of EVs is thoroughly considered. The effect of the across-time-and-space energy transmission on MMGS economic operation is analyzed to state the potential benefits of cooperative multi-objective optimization.

2. A cooperative multi-objective optimization model is established, including the dynamic economic dispatch of RESs, EVs, DC/AC converters, and the reactive power optimization of DC/AC converters in MMGS. The cooperative multi-objective optimization model consists of two loops. The inner-loop model uses the active power output of RESs, EVs, DC/AC converters as variables, and the daily operating cost of MMGS is used as the optimization objective. The outer-loop model uses the reactive power output of the DC/AC converters as the variable to optimize the

network loss of the distribution network, thereby reducing network loss cost and carbon emissions cost. The ultimate goal of the cooperative multi-objective is to obtain the optimal daily economic cost.

3. The concepts of carbon neutrality and carbon peaking are combined. Through the cooperative multi-objective optimization model, the carbon emissions generated by the operation of the MMGS and the distribution network are effectively reduced. The cooperative multi-objective optimization model not only improves the economy but also reduces the total carbon emissions of MMGS and the distribution network.

## 2. System Structure

### 2.1. Structure of the DC Multi-Microgrid System

The MMGS discussed in this paper includes multiple relatively independent MGs in space. The DC multi-microgrid energy management system (MMGEMS) manages all energy transactions in MMGS. Each MG is integrated into the distribution network through power electronic devices and exchanges energy with the distribution network. Each MG contains RESs and EVs charging/discharging infrastructures (EVCDIs). There are two main types of MGs in the MMGS: MGs located in residential areas (RMG) and MGs located in office areas (OBMG). The structure of the MMGS is shown in Figure 1.

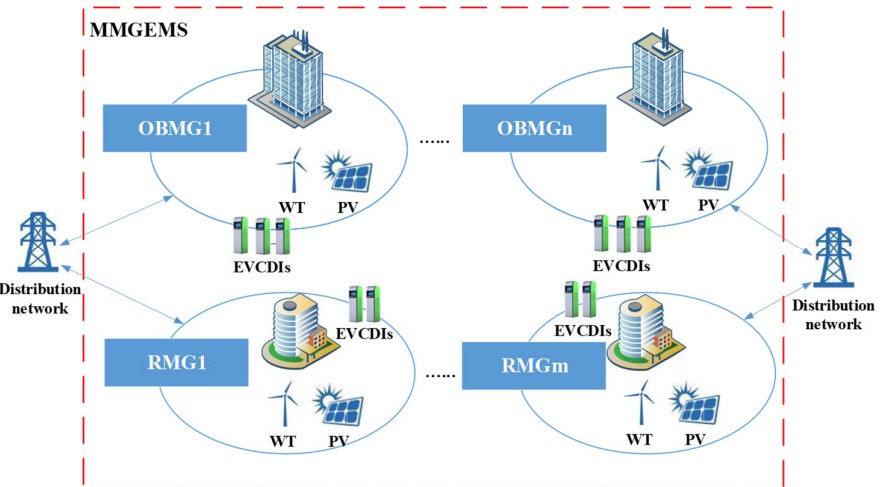

**Figure 1.** Structure of a DC multi-microgrid system.

The control of the system is mainly conducted by the collaboration of the MG energy management system (MEMS) and the EVs management system (EVMS). The MEMS is responsible for the energy dispatching of photovoltaics (PVs), wind turbines (WTs), and EVs in MGs, and the EVMS manages the charging and discharging behaviors of EVs. A DC multi-microgrid control system is shown in Figure 2.

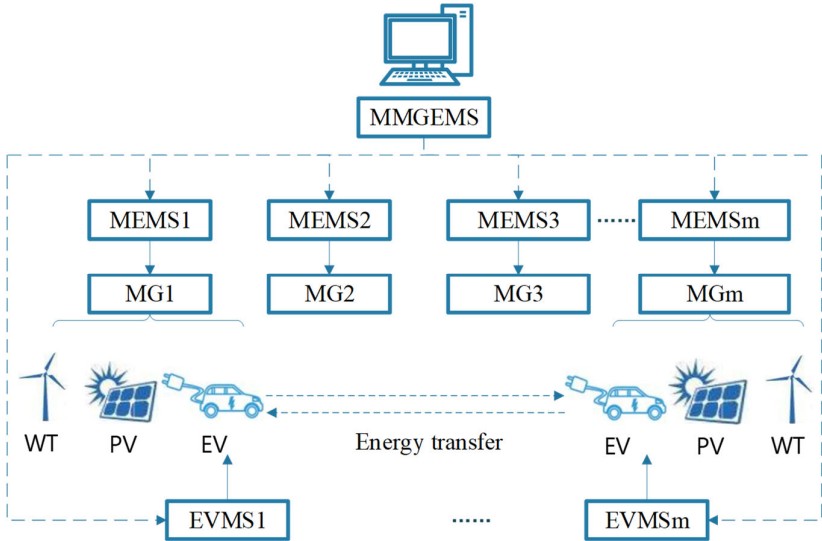

**Figure 2.** Structure of a DC multi-microgrid control system.

## 2.2. DC Microgrid

The basic structure of the DC microgrid is shown in Figure 3. Each MG is connected to the distribution network through a transformer and a DC/AC converter, which can exchange energy with the distribution network. A connection switch is installed in the grid-connected circuit, which can switch the MG between island operation mode and grid-connected operation mode.

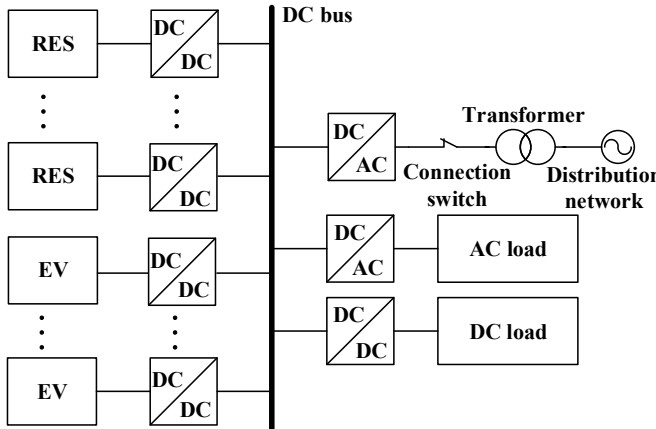

**Figure 3.** Structure of a DC microgrid.

## 2.3. Bidirectional DC/AC Converter

The bidirectional DC/AC converters are used to connect the MMGS and the distribution network, which can output active and reactive power with the distribution network. DC/AC converters use power factor correction (PFC) to obtain the unity power factor [21]. Therefore, the DC/AC converters are set to the unity power factor in this paper when the reactive power is not optimized [22]. However, by using the appropriate pulse-width modulation (PWM) switching technique, the power factor is adjusted to control the reactive power output of the DC/AC converters to the distribution network [22]. This is the basis for reactive power optimization.

## 3. Mathematical Model

### 3.1. Renewable Power Generation

3.1.1. Photovoltaic Module

In this paper, the power prediction module based on artificial neural networks (ANNs) [23] is applied to the economic dispatching of MMGS. The weather data are from the numerical weather forecast (NWP).

3.1.2. Wind Turbine

The output of WTs is mainly affected by wind speed [24]. The ANN is still used to predict wind power [25]. The inputs are the wind speed and wind direction from NWP.

### 3.2. Electric Vehicles

3.2.1. EVs Model

The EVs in the MMGS are all commuter vehicles, and the residents of the residential area are the workers in the office area. As 77.95% of EVs' users will reach the working area at 7:30–9:30 [26], the standard parking time slots in OBMG and RMG are assumed to be 9:00–17:00 and 19:00–7:00 [27,28]. The capacity of EV at $t$-th is

$$SOC_{EVm,n,t} = SOC_{EVm,n,t-1}(1-\sigma) + P_{m,n,t}^{EV} \bullet \Delta t \bullet \eta_{CEV} \, , \quad if \quad P_{m,n,t}^{EV} \geq 0 \tag{1}$$

$$SOC_{EVm,n,t} = SOC_{EVm,n,t-1}(1-\sigma) + P_{m,n,t}^{EV} \bullet \Delta t / \eta_{DEV} \, , \quad if \quad P_{m,n,t}^{EV} < 0 \tag{2}$$

The power output of EV in (1) and (2) is measured on the MMGS side. Where $SOC_{EVm,n,t}$ is the remaining power capacity of the $n$-th EV in the $m$-th MG in the $t$-th hour, $\sigma$ is the self-discharge coefficient. $P_{m,n,t}^{EV}$ is the charging or discharging power in the $t$-th hour of the $n$-th EV in the $m$-th MG. If $P_{m,n,t}^{EV} \geq 0$, EVs are charged. If $P_{m,n,t}^{EV} < 0$, EVs release energy; $\Delta t = 1$ h. $\eta_{DEV}$ and $\eta_{CEV}$ are the discharging and charging efficiency of EVs to calculate the power actual charging or discharging power of EVs.

3.2.2. Across-Time-and-Space Energy Transmission of EV

In the same MG, the EV is used as an energy storage unit, and its charging/discharging power can be dispatched for the operation of the MG. When the EV is connected to the MG and the power is sufficient, MMGS controls the EV to charge during the low charging price or when the system has excess energy, and discharge during the peak discharging price or when the system is short of power. The EV is charged and discharged in the same MG to realize energy transfer over time, thereby reducing the cost of MMGS purchasing electricity directly from the distribution network. At the same time, it also allows the user of the EV to profit by selling part of the electricity, which enables both parties to obtain a certain amount of economic benefit.

On the other hand, EVs not only have energy storage characteristics but also can move between different locations. In the case of differences in the electricity price of the distribution network within a region, benefits can be obtained through the cross-space transfer of energy. For example, the electricity prices of RMG and OBMG for electricity trading with the distribution network are quite different. Most of the time, the electricity price of OBMG purchasing electricity from the distribution network is higher than RMG. Therefore, the electric energy charged in the RMG at a low charging price is sold to the MMGS at a high discharging price in the OBMG, and the electric energy is transferred between different spaces and times through charging and discharging.

EVs realize the across-time energy transmission in the same MG and realize the across-time-and-space energy transmission in different MGs, which can transfer the lower-priced electric energy in RMG to OBMG at a higher price. Under the right conditions, both MMGS and EV users can benefit. This characteristic of EVs for energy transfer between different times and different spaces is called the across-time-and-space energy transmission.

*3.3. EV Charging/Discharging Infrastructures*

The charging/discharging behaviors of EVs are carried out through the EVCDIs.

$$P_{m,n,t}^{EVCDIs} = P_{m,n,t}^{EV} \bullet \eta_{CEV} \ , \quad if \quad P_{m,n,t}^{EV} \geq 0 \tag{3}$$

$$P_{m,n,t}^{EVCDIs} = P_{m,n,t}^{EV} / \eta_{DEV} \ , \quad if \quad P_{m,n,t}^{EV} < 0 \tag{4}$$

where $P_{m,n,t}^{EVCDIs}$ is the power of the EVCDIs of the *n*-th EV in the *m*-th MG in the *t*-th hour.

## 4. Cooperative Multi-Objective Optimization Model

*4.1. Description of the Optimization Model*

The EVMS collects the dispatchable capacity forecast data of EVs and outputs the dispatching plan of the EVs. The MEMS collects the output of predicted RESs, the predicted load data, and the energy price of the distribution network. Based on this information, MEMS outputs the active power of RESs, EVs, and DC/AC converters in MMGS, and transmits it to the reactive power optimization module in MMGEMS. The reactive power optimization module outputs the optimal reactive power of the DC/AC converters according to the data. The two modules coordinate and output the optimal result.

*4.2. Double-Loop Optimization Process*

The process is shown in Figure 4. The inner-loop is the dynamic economic dispatch which is used to optimize the active power output of RESs, EVs, and DC/AC converters to obtain the optimal total operating cost of the MMGS. The outer-loop optimizes the reactive power output of the DC/AC converters according to the active power output of the inner-loop, to make the network loss of the distribution network minimum, thereby reducing the network loss cost and carbon emissions of MMGS and the distribution network. The inner-loop and the outer-loop work together to obtain the optimal active power output plan in MMGS and reactive power output of the DC/AC converters, which makes the economic cost of MMGS minimum.

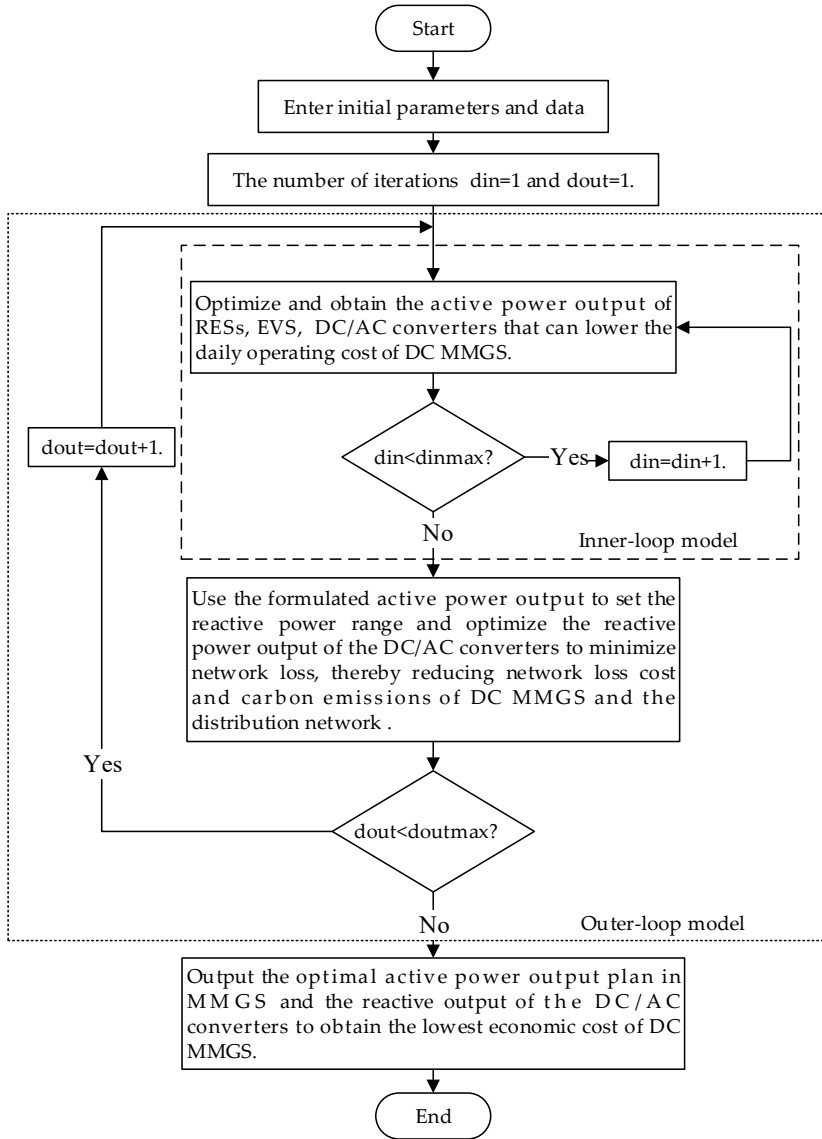

**Figure 4.** Process of the cooperative multi-objective optimization.

### 4.3. Cooperative Multi-Objective Optimization Objective Function

The main goal of optimization is to reduce the daily economic total cost of MMGS. MMGS discussed in this model consists of multiple MGs, which are assumed to be owned by a single operator. Another goal of the model is the lowest network loss of the distribution network, which can be obtained through the outer-loop model. Therefore, the objective function to minimize the total economic cost of the entire system can be expressed as:

$$f = C_{ETC} = C_{OTC} + C_{WTC} \tag{5}$$

where $f$ is the main goal of the cooperative optimization, $C_{ETC}$ is the economic total cost of MMGS. $C_{OTC}$ is the operating total cost of MMGS, $C_{WTC}$ is the energy loss cost of the MMGS that is obtained from the outer-loop model.
where

$$C_{WTC} = C_{il} + C_{co} \tag{6}$$

$$C_{co} = E_{CO} \bullet k_c \tag{7}$$

$$E_{CO} = (W_S^G - W_S^B) \bullet \Delta t \bullet e_c \tag{8}$$

$$C_{il} = (W_S^G - W_S^B) \bullet \Delta t \bullet k_{il} \tag{9}$$

$$E_C = \sum_{m=1}^M E_{CIm} + E_{CO} \tag{10}$$

$C_{il}$ and $C_{co}$ are the network loss cost and carbon emissions cost caused by the increase in the distribution network loss in the outer-loop model, respectively. $E_{co}$ is the carbon emissions generated by the distribution network. $W_S^G$ is the total daily operating network loss of the distribution network when MMGS is integrated into the distribution network and runs. $W_S^B$ is the total daily operating network loss when there is no MMGS access, which is a fixed value also called the original baseline loss. $k_{il}$, $e_c$, $k_c$ are fixed factors, $k_{il}$ is the loss cost coefficient of the distribution network, $e_c$ is the carbon emissions factor, $k_c$ is the carbon cost factor. $\Delta t = 1$ h. $E_c$ is the total carbon emissions of MMGS and the distribution network, $E_{CIm}$ is the carbon emissions generated by $m$-th MG in the inner-loop model, $M$ is the number of MGs in the MMGS.

Since $C_{OTC}$ and $W_S^G$ are the optimization targets of the inner-loop model and the outer-loop model, respectively, the objective functions of the inner-loop model and the outer-loop model are set as follows:

$$f_1 = \min C_{OTC} \tag{11}$$

$$f_2 = \min W_S^G \tag{12}$$

where $f_1$ and $f_2$ are the objective functions of the inner-loop model and the outer-loop model, respectively.

Through (5)–(12), $f$ can be expressed as:

$$f = f_1 + (f_2 - W_S^B) \bullet \Delta t \bullet \left[ e_c \bullet k_c + k_{il} \right] \tag{13}$$

*4.4. Inner-Loop Optimization*

The goal of the inner-loop optimization model is to minimize the daily operating cost of the MMGS. The daily operating cost is mainly composed of system energy transaction cost and carbon emissions cost. The objective function is as follows:

$$f_1 = \min C_{OTC} \tag{14}$$

$$C_{OTC} = \sum_{m=1}^M C_{OCm} \tag{15}$$

$$C_{OCm} = C_{exm} + C_{cim} \tag{16}$$

$$C_{cim} = E_{CIm} \bullet k_c \tag{17}$$

$C_{OCm}$ is the operating cost of the $m$-th MG that is obtained from the inner-loop model. $C_{exm}$ is the energy transaction cost in the $m$-th MG. $C_{cim}$ is the carbon emissions cost due to energy exchange in the inner-loop model.

4.4.1. Energy Transaction Cost

The energy transaction cost is the sum of RESs cost, energy exchange cost between MMGS and EVs, MMGS and distribution network, and the additional cycle cost of EV batteries. $P_{m,t}^{EV}$, $P_{m,t}^{PV}$, $P_{m,t}^{WT}$, and $P_{m,t}^G$ are the optimization variables.

$$C_{exm} = C_{resm} - C_{evm} + C_{gm} + C_{cym} \tag{18}$$

$$C_{resm} = C_{PVm} + C_{WTm} \tag{19}$$

$$C_{PVm} + C_{WTm} = \sum_{t=1}^{T} P_{m,t}^{PV} C_{m,t}^{PV} \Delta t + \sum_{t=1}^{T} P_{m,t}^{WT} C_{m,t}^{WT} \Delta t \tag{20}$$

$$C_{evm} = \sum_{t=1}^{T} \sum_{n=1}^{N} P_{m,n,t}^{EV} \bullet \eta_{CEV} C_{m,t}^{CEV} \Delta t, \quad if \quad P_{m,n,t}^{EV} \geq 0 \tag{21}$$

$$C_{evm} = \sum_{t=1}^{T} \sum_{n=1}^{N} P_{m,n,t}^{EV} C_{m,t}^{DEV} \Delta t, \quad if \quad P_{m,n,t}^{EV} < 0 \tag{22}$$

$$C_{gm} = \sum_{t=1}^{T} P_{m,t}^{G} C_{m,t}^{G} \Delta t \tag{23}$$

$$C_{cym} = \sum_{n=1}^{N} k_{cy} C_{cyn}^{EV} \tag{24}$$

where $C_{resm}$ is the cost of RESs of the $m$-th MG in a day, $C_{PVm}$, and $C_{WTm}$ are the cost of PVs and WTs. $P_{m,t}^{PV}$ is the power output of PVs in the $m$-th MG, at $t$-th hour, $C_{m,t}^{PV}$ is the PV power generation cost, $P_{m,t}^{WT}$ is the power output of WTs, $C_{m,t}^{WT}$ is the WT power generation cost. $C_{evm}$ is the cost of energy exchange between MMGS and EVs, $C_{m,t}^{CEV}$ is the charging price of EVs in $m$-th MG, $C_{m,t}^{DEV}$ is the discharging price, $\Delta t$ = 1 h, $T$ = 24 h. $C_{gm}$ is the energy exchange cost between the MG and the distribution network through the DC/AC converters, $P_{m,t}^{G}$ is the active power output between the MG and the distribution network through the DC/AC converters. If $P_{m,t}^{G} \geq 0$, MG purchases electricity from the distribution network. If $P_{m,t}^{G} < 0$, MMGS sells electricity to the distribution network. $C_{m,t}^{G}$ is the electricity price that MG purchases/sells to the distribution network. $C_{cym}$ is the additional cycle cost of EV batteries, $C_{cyn}^{EV}$ is the additional battery charging/discharging cycle cost of $n$-th EV, $k_{cy}$ is the number of additional charging/discharging cycles, $N$ is the number of EVs.

### 4.4.2. Carbon Emissions and Cost

The electricity of the distribution network mainly comes from thermal power generation. When MG exchanges energy with the distribution network, the distribution network emits more $CO_2$. To reduce carbon emissions as much as possible and increase the use of RESs, in this paper, the cost of carbon emissions is used as the penalty cost of $CO_2$ generated by the energy exchange between the MMGS and the distribution network [29].

$$C_{cim} = E_{CIm} \bullet k_c \tag{25}$$

$$E_{CIm} = \sum_{t=1}^{T} P_{m,t}^{G} \Delta t \bullet e_c \tag{26}$$

### 4.5. Constraints of the Inner-Loop Model

#### 4.5.1. EVs Power Constraint

The charging/discharging power of EVs cannot exceed the rated power of EVCDIs.

$$\left| P_{m,n,t}^{EV} \right| \leq P_{m,n,R}^{EVCDIs} \tag{27}$$

where $P_{m,n,R}^{EVCDIs}$ is the rated power of the EVCDI serving the $n$-th EV.

### 4.5.2. EVs Capacity Constraint

The remaining power of EVs must meet the constraints of rated capacity.

$$SOC_{EVm,n,\min} \leq SOC_{EVm,n,t} \leq SOC_{EVm,n,\max} \tag{28}$$

where $SOC_{EVm,n,min}$ and $SOC_{EVm,n,max}$ are the minima and maximum capacity, respectively, of $n$-th EV in $m$-th MG.

### 4.5.3. RESs Output Constraint

Considering the performance limitations of renewable energy, the output of RESs in $m$-th MG has a certain upper limit.

$$0 \leq P_{m,t}^{WT} \leq P_{m,\max}^{WT} \tag{29}$$

$$0 \leq P_{m,t}^{PV} \leq P_{m,\max}^{PV} \tag{30}$$

### 4.5.4. System Power Balance Constraint

For MMGS, the active power output should meet the power balance constraint.

$$P_{m,t}^{EV} + P_{m,t}^{G} + P_{m,t}^{WT} + P_{m,t}^{PV} = P_{m,t}^{L} \tag{31}$$

where $P_{m,t}^{L}$ is the total load of the $m$-th MG at time $t$.

### *4.6. Outer-Loop Optimization*

The outer-loop optimization model takes the network loss as the optimization goal. By optimizing the reactive power output of the DC/AC converters $Q_{m,t}^{G}$, the daily network loss of the distribution network is minimized, thereby reducing network loss cost and carbon emissions of MMGS and the distribution network [30]. This paper assumes that the $m$-th MG is connected to node $i$ of the distribution network.

$$f_2 = \min W_S^G \tag{32}$$

$$W_S^G = \sum_{t=1}^{T} \sum_{i,j=1}^{N_{br}} k_i R_{ij} \frac{P_{ij,t}^2 + Q_{ij,t}^2}{V_{ij,t}^2} \tag{33}$$

$$W_S^I = W_S^G - W_S^B \tag{34}$$

$$P_{ij,t} = P_{ij,t}^0 + P_{m,t}^G \tag{35}$$

$$Q_{ij,t} = Q_{ij,t}^0 + Q_{m,t}^G \tag{36}$$

$$W_S^I = f_2 - W_S^B \tag{37}$$

$$C_{il} + C_{co} = W_S^I \bullet \Delta t \bullet k_{il} + W_S^I \bullet \Delta t \bullet e_c \bullet k_c \tag{38}$$

$W_S^I$ is the daily operating increased network loss of the distribution network. $N_{br}$ is the number of branches. $i$, $j$ are the nodes, $k_i$ is the state variable of the $i$-th branch switch, 1 means closed, 0 means open; $R_{ij}$ is the resistance of branch $ij$, $P_{ij,t}$, $Q_{ij,t}$ are the active and reactive power of branch $ij$ in the $t$-th hour, $V_{ij,t}$ is the voltage, $P_{ij,t}^0$, $Q_{ij,t}^0$ are initially active, reactive power when connected without MMGS. $P_{m,t}^G$, $Q_{m,t}^G$ are the active and reactive power through the DC/AC converters injected into node $i$ by the $m$-th MG connected to node $i$. To facilitate the calculation of network loss, a day is divided into 12 small periods, with a time interval of 2 h.

### 4.6.1. Network Loss Cost

The operation of MMGS connected to the distribution network will cause increased network loss in the distribution network. Therefore, the distribution network will sign a

contract with the operator of MMGS, and the operator needs to pay a certain network loss fee for the daily operating increased network loss in the distribution network.

$$C_{il} = W_S^I \bullet \Delta t \bullet k_{il} \tag{39}$$

### 4.6.2. Carbon Emissions and Cost

When the network loss of the distribution network increases by the operation of MMGS, more $CO_2$ will be emitted. MMGS will still incur a penalty cost for carbon emissions by the increasing network loss, which differs from the carbon emissions cost due to energy exchange in the inner-loop model.

$$C_{co} = W_S^I \bullet \Delta t \bullet e_c \bullet k_c \tag{40}$$

### 4.7. Constraints of the Outer-Loop Model

The model takes the actual power flow of the power grid as the constraints.

#### 4.7.1. Node Power Flow Constraint

$$P_{m,t}^G + P_{i,t}^0 = P_{Li,t} + V_{i,t} \sum_{j=1}^{N_n} V_{j,t}(G_{ij}\cos\delta_{ij} + B_{ij}\sin\delta_{ij}) \tag{41}$$

$$Q_{m,t}^G + Q_{i,t}^0 = Q_{Li,t} + V_{i,t} \sum_{j=1}^{N_n} V_{j,t}(G_{ij}\sin\delta_{ij} + B_{ij}\cos\delta_{ij}) \tag{42}$$

where $P_{i,t}^0$ and $Q_{i,t}^0$ are the initial input active and reactive power of node $i$ in the $t$-th hour, $P_{Li,t}$ and $Q_{Li,t}$ are the active and reactive load, $V_{i,t}$ and $V_{j,t}$ are the voltage of node $i$ *and j*, $G_{ij}$, $B_{ij}$, and $\delta_{ij}$ are the conductance, susceptance, and phase angle difference of branch $ij$.

#### 4.7.2. Node Voltage Constraint

$$V_i^{\min} \le V_{i,t} \le V_i^{\max} \tag{43}$$

$V_i^{min}$ and $V_i^{max}$ are lower and upper limits of the node $i$ voltage amplitude.

#### 4.7.3. Branch Power Constraint

$$\left|P_{ij,t}\right| \le P_{ij,\max} \tag{44}$$

$$\left|Q_{ij,t}\right| \le Q_{ij,\max} \tag{45}$$

$P_{ij,max}$, $Q_{ij,max}$ are the maximum active and reactive power of the branch $ij$.

#### 4.7.4. Branch Current Constraint

$$I_{ij} \le I_{ij}^{\max} \tag{46}$$

where $I_{ij}^{max}$ is the upper limit of branch $ij$ current carrying capacity.

#### 4.7.5. Reactive Output Constraint of DC/AC Converter

The reactive power output of the DC/AC converters must satisfy the constraint:

$$\left|Q_{m,t}^G\right| \le \sqrt{S_m^2 - (P_{m,t}^G)^2} \tag{47}$$

where $S_m$ is the rated power of the DC/AC converter in $m$-th MG, $Q_{m,t}^G$ is the reactive power that the DC/AC converter can output to the distribution network, $P_{m,t}^G$ is the active power output by DC/AC converter.

### 4.8. Particle Swarm Algorithm

#### 4.8.1. Procedure of PSO

The steps of particle swarm optimization (PSO) are ~~as follows~~ shown in Figure 5 [31].

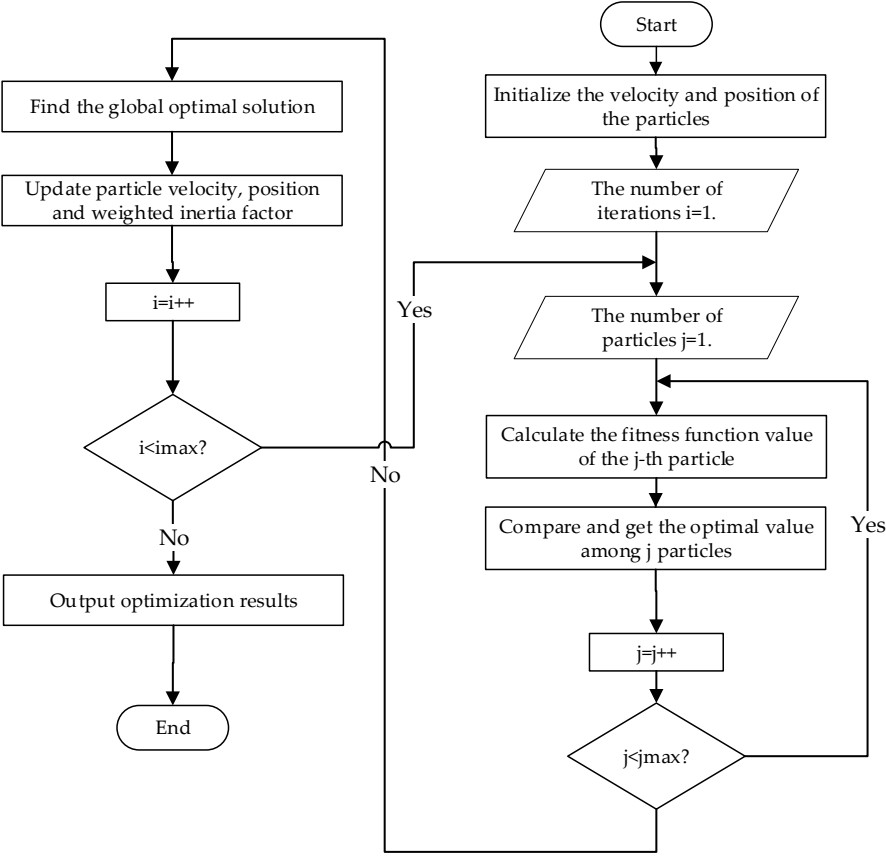

**Figure 5.** Process of the PSO algorithm.

#### 4.8.2. Coding

In the inner-loop, the coding about the economic dispatch of RESs, EVs, and DC/AC converters can be represented by a real-valued matrix. $k$ is the index of the particle of the inner-loop. $M$ is the number of MG. $T$ is the dispatching cycle.

$$I_{\mathrm{MG}}^{k} = \begin{bmatrix} I_{\mathrm{MG1}} \\ I_{\mathrm{MG2}} \\ \vdots \\ I_{\mathrm{MGm}} \\ \vdots \\ I_{\mathrm{MGM}} \end{bmatrix} \tag{48}$$

$$I_{\mathrm{MGm}} = \begin{bmatrix} P_{m,1}^{PV} & P_{m,2}^{PV} & \cdots & P_{m,t}^{PV} & \cdots & P_{m,T}^{PV} \\ P_{m,1}^{WT} & P_{m,2}^{WT} & \cdots & P_{m,t}^{WT} & \cdots & P_{m,T}^{WT} \\ P_{m,1}^{G} & P_{m,2}^{G} & \cdots & P_{m,t}^{G} & \cdots & P_{m,T}^{G} \\ P_{m,1}^{EV} & P_{m,2}^{EV} & \cdots & P_{m,t}^{EV} & \cdots & P_{m,T}^{EV} \end{bmatrix} \tag{49}$$

$P_{m,t}^{EV}$, $P_{m,t}^{PV}$, $P_{m,t}^{WT}$, and $P_{m,t}^{G}$ are the power outputs of EVs, PVs, WTs, and DC/AC converters in the $m$-th MG in the $t$-th hour, respectively.

In the outer-loop, the coding about the reactive power output by DC/AC converters can be represented by a real-valued matrix. *s* is the index of the particle of the outer-loop.

$$O_{MG}^{s} = \begin{bmatrix} O_{MG1} \\ O_{MG2} \\ \vdots \\ O_{MGm} \\ \vdots \\ O_{MGM} \end{bmatrix} \tag{50}$$

$$O_{MGm}^{s} = \begin{bmatrix} Q_{m,1}^{G} & Q_{m,2}^{G} & \cdots & Q_{m,t}^{G} & \cdots & Q_{m,T}^{G} \end{bmatrix} \tag{51}$$

$Q_{m,t}^{G}$ is the reactive power output of DC/AC converters in the *m*-th MG at *t*-th hour.

However, the dispatch range of the outer-loop variable also changes when the variable of the DC/AC converters changes in the inner-loop. Therefore, a dynamic range adjustment algorithm is added to the outer-loop model.

$$\left| Q_{m,t}^{G} \right|^{max} = \sqrt{S_{m}^{2} - (P_{m,t}^{G})^{2}} \tag{52}$$

The inner-loop and outer-loop cooperate to generate the optimal optimization results.

## 5. Case Study and Discussion

### 5.1. Case Description

There are 30 EVs concentrated in OBMG/RMG for the charging/discharging service [31]. The dispatching cycle is 24 h. This paper sets up four cases to analyze the optimization model. By using NWPs from Wuhan City, Hubei Province, China in June 2020, a day's renewable power generation in summer is predicted as the input of the model.

#### 5.1.1. Case 1

In this case, the EVs do not participate in the energy dispatch of the MG. Once they reach the MG, the EVs will be charged until the batteries are fully charged. MMGS does not optimize the reactive power output by DC/AC converters.

#### 5.1.2. Case 2

In this case, after EVs are connected to the MG, they participate in the energy management system of each MG. Once they reach the MG, the energy in the EV battery will be dispatched by the MG's energy management system until they leave. When the EVs leave the MG at the end of the dispatching, the energy of EVs should be fully charged. This case takes advantage of the across-time energy transmission of EVs in each independent MG, and the optimization of the reactive power output of DC/AC converters is not considered.

#### 5.1.3. Case 3

In this case, only the inner-loop economic dispatch model is used to minimize the total cost of MMGS by optimizing the active power output of RESs, EVs, and DC/AC converters. The ATSET of EV between RMGs and OBMGs is used. However, the reactive power output of DC/AC converters is also not optimized. Case 3 can be used as a reference.

#### 5.1.4. Case 4

In this case, cooperative multi-objective optimization combines the inner-loop economic dispatch model and the outer-loop reactive power optimization model. The ATSET of EV between RMG and OBMG is considered. By optimizing the active power output of RES, EVs, and DC/AC converters, the total daily operating cost of MMGS is reduced. By optimizing the reactive power output of DC/AC converters, the loss of the distribution network is reduced, and the total economic cost of MMGS is reduced synergistically.

### 5.2. Simulation System Construction

#### 5.2.1. System Introduction

The modified IEEE 33-node system is used to prove the model, whose structure is shown in Figure 6, and its parameters can be obtained from [32]. According to the principle of distribution [32], OBMG and RMG are set at node 19 and node 20, respectively.

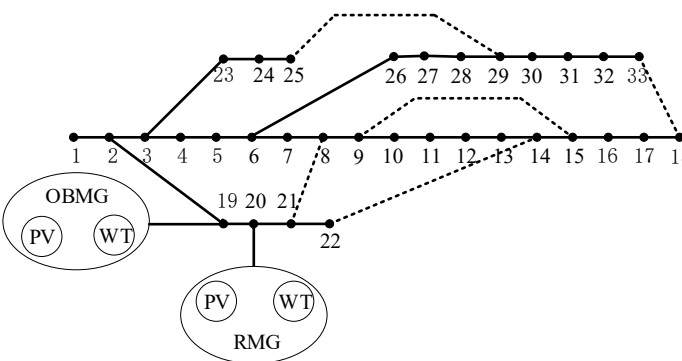

**Figure 6.** Topological diagram of the modified IEEE 33-node system.

#### 5.2.2. Parameters of RESs

According to the principle of renewable energy consumption [27], the RESs installed in each MG and the power generation cost are given in Table 1. The optimization time interval is 1 h, and the optimization cycle is 1 day (24 h).

**Table 1.** Installed RESs and the power generation cost.

| MG Type | RES Type | Installed Capacity/kW | Power Generation Cost/¥·kWh⁻¹ |
|---------|----------|----------------------|-------------------------------|
| OBMG    | PV1      | 800                  | 0.24                          |
|         | WT1      | 800                  | 0.38                          |
| RMG     | PV2      | 400                  | 0.24                          |
|         | WT2      | 400                  | 0.38                          |

The daily wind speed, radiation intensity, temperature, and load data are adopted in this area. The renewable energy output and load curves of each MG come from [27].

#### 5.2.3. Parameters of DC/AC Converter

Considering the performance of the DC/AC converters of MMGS, $S_m$ = 1000 kW, the power limit is set as [33]:

$$0 \le \left| P_{m,t}^G \right| \le 1000 \text{kW} \tag{53}$$

$$0 \le \left| Q_{m,t}^G \right| \le 1000 \text{kVar} \tag{54}$$

### 5.2.4. Parameters of EVs

Take a BYD E6 electric vehicle as an example, whose parameters are from [34]. The battery capacity is 80 kWh, and the upper limit of charging and discharging power of EVCDI is 7 kW. The charging and discharging efficiency are all 90% [34]. An EV consumes an average of 8% of electricity per way between RMG and OBMG [31]. The additional battery charging/discharging cycle cost of EV is CNY 50 each time [35]. The minimum power of the battery of EV is not less than 20% [36]. Considering the needs of users, the upper and lower limits for the battery are 100% and 35% [27].

### 5.2.5. Other Parameters

The time-of-use (TOU) energy prices in RMG/OBMG from [31] are shown in Figure 7. The carbon emissions factor $e_c$ is 86.47 g/kWh [29], and the carbon cost factor $k_c$ is 0.21 CNY/kg [37]. The loss cost coefficient of the distribution network $k_{il}$ is 0.74 CNY/kWh [38].

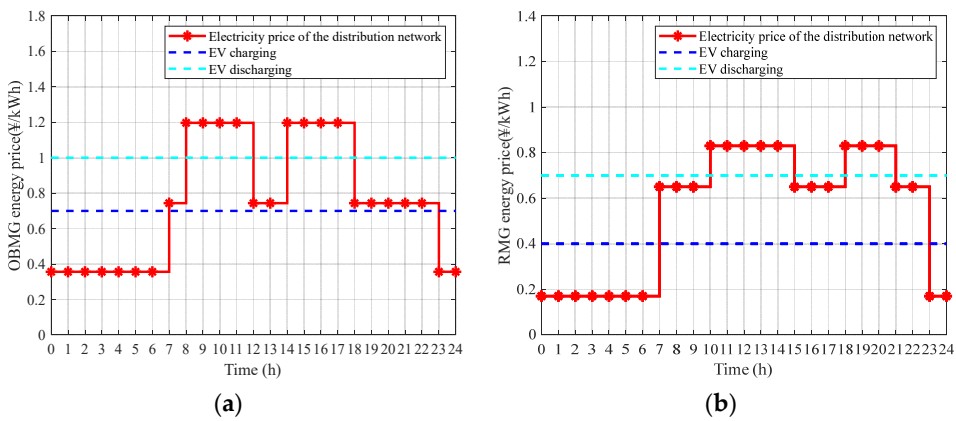

**Figure 7.** (**a**) Prices of energy exchanging in OBMG; (**b**) prices of energy exchanging in RMG.

### *5.3. Simulation Results*

### 5.3.1. Inner-Loop Optimization Results

1.   Case 1

In this case, when EVs are connected to the MG, they are charged immediately. In case 1, the 24 h curve of RESs, EVs, load, and DC/AC converter active power output in OBMG/RMG is shown in Figure 8. EVs are charged as soon as they reach RMG/OBMG. The active power curve of the DC/AC converter represents the active power output curve of the MG to the distribution network. When it is below the *X*-axis, it means that the MG sells electric energy to the distribution network. When it is above the *X*-axis, it means that the MG purchases electric energy from the distribution network. The power curve of EVs has a similar definition. In Figure 8, RMG will allow EVs to be charged at maximum power from 19:00–20:00, and when RESs are insufficient, MEMS will purchase electricity from the distribution network. OBMG is also charging EVs at 9:00–10:00. The total daily operating cost of RMG is CNY 2776.3, and the total daily operating cost of OBMG is CNY 5732.6. Therefore, the total daily operating cost of MMGS is CNY 8508.9.

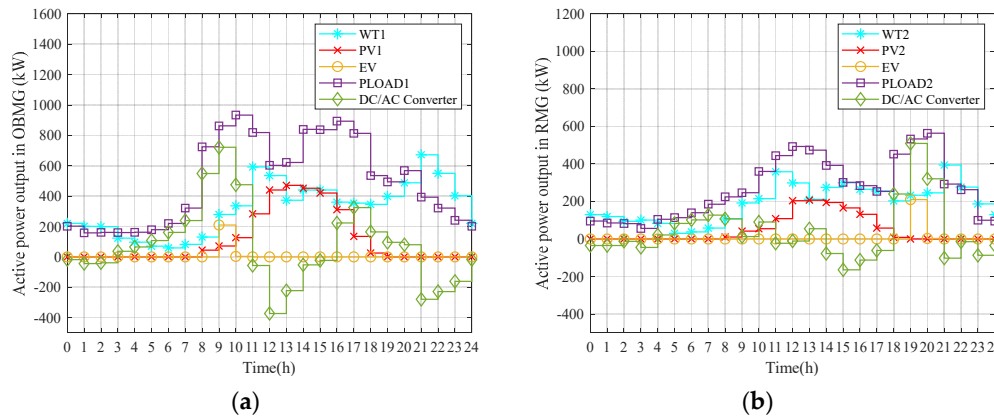

**Figure 8.** (**a**) Power output of RESs, EVs, and the DC/AC converter in OBMG; (**b**) power output of RESs, EVs, and the DC/AC converter in RMG.

2. Case 2

In this case, since EVs can participate in the energy dispatching of independent MGs, their across-time energy transmission is used. When the total generated power of the RESs in the MGs is greater than the load, the MEMS will sell the remaining energy to the distribution network or charge the EVs according to the energy prices. When the total power generation of RES is less than the load, the MEMS will purchase electricity from the distribution network or EVs according to the energy prices. In Figure 9, the active power output of RES, EV and DC/AC converters in OBMG and MG are optimized. In OBMG, due to the high energy prices of the distribution network and EVs from 9:00 to 12:00, MEMS choose to let EVs release electric energy. OBMG lowers costs by selling energy to the distribution network. When energy prices are low between 12:00 and 15:00, MEMS fully charges EVs. In RMG, MEMS chooses to charge EVs at 23:00 when energy prices are low. This is to avoid additional battery charge–discharge cycle costs due to discharge, so EVs are only charged. The across-time energy transmission of the EV in the independent MG is fully utilized. Through optimization model calculation, the total daily operating cost of RMG is CNY 2644.1, and the total daily operating cost of OBMG is CNY 5642.1. Therefore, the total daily operating cost of MMGS is CNY 8286.2. Compared with Case 1, the across-time energy transmission of EVs can reduce the overall operating cost of MMGS.

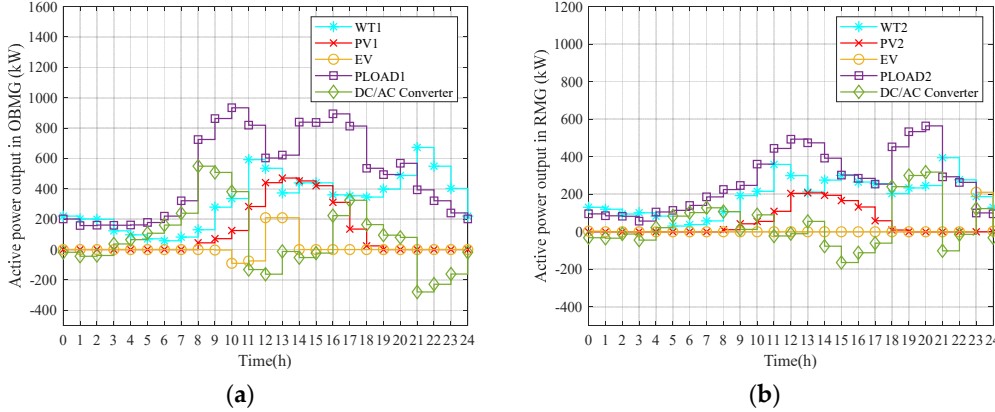

**Figure 9.** (**a**) Power output of RESs, EVs, and the DC/AC converter in OBMG; (**b**) power output of RESs, EVs, and the DC/AC converter in RMG.

3. Case 3

In this case, EVs can transfer energy among multiple MGs, and EVs participate in MMGS energy dispatching. For OBMG, the energy price of the distribution network and the discharging price of EVs are both high, and the difference between the energy price of the distribution network and the discharging price of EV is much higher than that of RMG. Therefore, MMGS's energy management system will discharge almost all EVs as much as possible when EVs are connected to OBMG, and earn more profits. For RMG, its advantage is that the charging price is lower, so MMGEMS will try its best to allow almost all EVs to be charged during the low energy price of RMG to reduce the charging cost of EVs. In Figure 10, EVs are discharged as much as possible in OBMG and then charged as much as possible in RMG. After optimization model calculation, the total daily operating cost of RMG is CNY 2391.8, and the total daily operating cost of OBMG is CNY 5404.1. Therefore, the total daily operating cost of MMGS is CNY 7795.9. However, compared with case 1 case 2, by using the across-time-and-space energy transmission of EVs, the total daily operating cost of the MMGS is the lowest in this case.

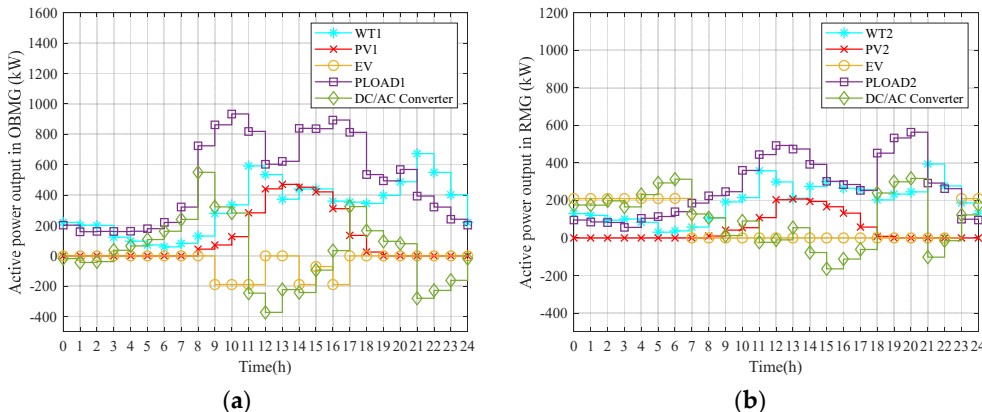

**Figure 10.** (**a**) Power output of RESs, EVs, and the DC/AC converter in OBMG; (**b**) power output of RESs, EVs, and the DC/AC converter in RMG.

Table 2 is the comparison of the results of the inner-loop economic dispatch in the three cases. Since case 4 and case 3 use the same inner-loop economic dispatch model, their inner-loop output conditions are the same. Here, the effectiveness of the inner-loop economic dispatch model is mainly discussed, so there is no need to show the results in case 4.

In case 1, EVs do not participate in the energy dispatching of the MG, and MMGS has the highest total operating cost. In case 2, the across-time energy transmission of EVs in the independent MG is used to reduce the cost. In case 3 and case 4, the across-time-and-space energy transmission of EVs is considered to further reduce the total daily operating cost of MMGS, which achieves the lowest daily operating cost $C_{OTC}$.

**Table 2.** Comparison of operating cost in three cases.

| Case | MGs | $C_{ex}$ Energy Transaction Cost/CNY | $C_{ci}$ Carbon Emissions Cost/CNY | $C_{OTC}$ Total Operating Cost/CNY |
|------|------|------|------|------|
| Case 1 | RMG | 2760.0 | 16.3 | 2776.3 |
| | OBMG | 5700.8 | 31.8 | 5732.6 |
| | MMGS | 8460.8 | 48.1 | 8508.9 |
| Case 2 | RMG | 2627.8 | 16.3 | 2644.1 |
| | OBMG | 5609.6 | 32.5 | 5642.1 |
| | MMGS | 8237.4 | 48.8 | 8286.2 |
| Case 3 | RMG | 2348.9 | 42.9 | 2391.8 |
| | OBMG | 5394.6 | 9.5 | 5404.1 |
| | MMGS | 7743.5 | 52.4 | 7795.9 |

Figure 11 is the remaining capacity curve of EVs, which proves that EVs meet the power constraint in the four cases. It is also verified that the charging and discharging behaviors analyses of EVs in the three cases are correct.

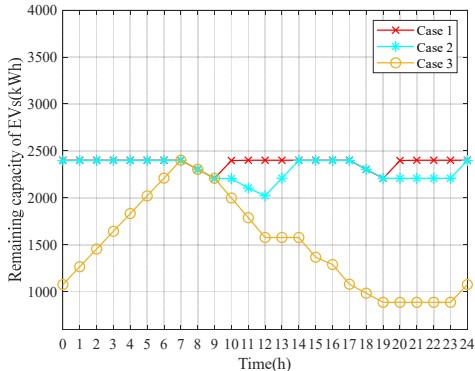

**Figure 11.** Remaining capacity of EVs in three cases.

Table 3 is the cost of EVs' users. Among the three cases, the user cost of case 3 is the lowest. In case 1, EVs do not participate in dispatching, and the cost of users is the highest. In case 2, the cost of users is reduced by the across-time energy transmission. In case 3, the inner-loop economic dispatch is adopted, which makes full use of the across-time-and-space energy transmission of EVs. Additionally, the cost of users is further reduced. Combining with the lowest daily operating cost of MMGS, the inner-loop economic dispatch model using ATSET of EVs achieved a win–win situation for MMGS and EVs' users.

**Table 3.** The cost of EVs' users in three cases in one day.

| Case | Case 1 | Case 2 | Case 3 |
|------|------|------|------|
| The Cost of EVs' Users/CNY | 211.2 | 173.4 | −410.8 |

5.3.2. Outer-Loop Optimization Results

In case 1, case 2, and case 3, the reactive power output of the DC/AC converters is not optimized. In case 4, the outer-loop reactive power optimization model is used to optimize the reactive power output of the DC/AC converters. The optimized reactive power output of the DC/AC converters of RMG and OBMG in case 4 is shown in Figure 12. The converters will absorb or output a certain amount of reactive power to the distribution

network at every moment, which is used to optimize the operating network loss of the distribution network, thereby reducing the energy loss cost $C_{WTC}$ and total carbon emissions $E_C$ of MMGS and the distribution network, and cooperating with the inner-loop model to reduce the total economic cost $C_{ETC}$ of MMGS. The comparison of the results under the four cases is shown in Table 4.

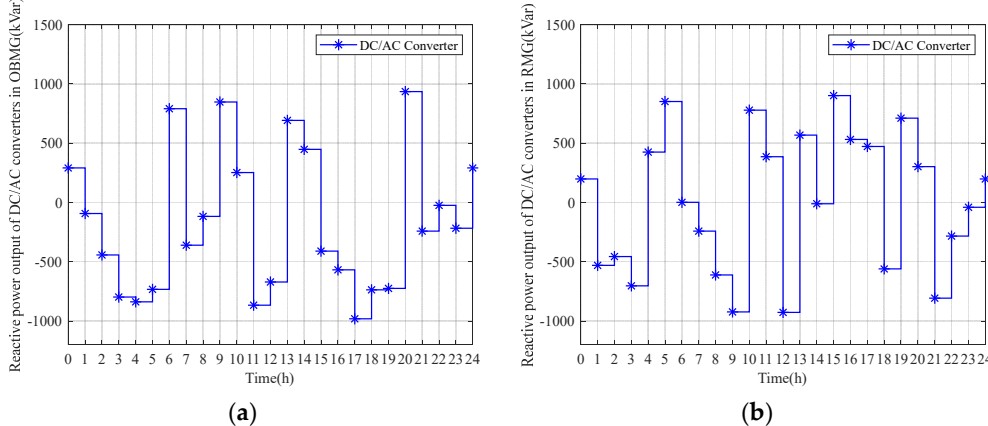

**Figure 12.** (**a**) Reactive power output by the DC/AC converter of OBMG in case 4; (**b**) reactive power output by the DC/AC converter of RMG in case 4.

**Table 4.** Comparison of network loss in the four cases.

| Case | Case 1 | Case 2 | Case 3 | Case 4 |
|---|---|---|---|---|
| $W_S^G$ Total Network Loss/kW | 14,889.2 | 14,872.5 | 14,876.0 | 13,987.2 |
| $W_S^I$ Increased Network Loss/kW | 101.3 | 84.6 | 88.1 | −800.7 |

By analyzing the distribution network loss under the above different cases, it can be concluded that the reactive power output of the DC/AC converters to the distribution network will affect the distribution network loss. When MMGS is not integrated into the distribution network to work, the original baseline loss $W_S^B$ of the distribution network is 14,787.9 kW. The distribution network loss under the first three cases is all greater than $W_S^B$, while the distribution network loss under case 4 is less than $W_S^B$ and lower than the first three cases. Case 3 and case 4 are a set of comparisons. Under the common premise of using the inner-loop optimization model, case 4 that uses reactive power optimization has lower network loss. Figure 13 is the increased network loss diagram for each period of the distribution network which further proves that intelligently optimizing the reactive power output of DC/AC converters through the outer-loop model can effectively reduce the daily network loss of the distribution network.

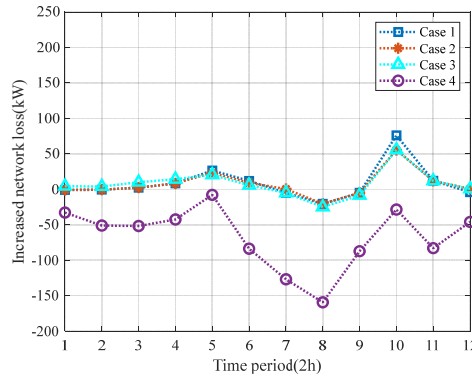

**Figure 13.** Increased network loss in four cases.

Table 5 is the network loss cost and energy loss cost in four cases. Among the four cases, the network loss cost $C_{il}$ and the carbon emissions cost $C_{co}$ derived from the optimization of the outer-loop model are the lowest, which proves that the outer-loop optimization model plays a role in the cooperative optimization of the economic cost of MMGS.

**Table 5.** Network loss cost and energy loss cost in the four cases.

| Case | $C_{il}$ | $C_{co}$ | $C_{WTC}$ |
|---|---|---|---|
| | Network Loss Cost/CNY | Carbon Emissions Cost/CNY | Energy Loss Cost/CNY |
| Case 1 | 75.0 | 1.8 | 76.8 |
| Case 2 | 62.6 | 1.5 | 64.1 |
| Case 3 | 65.2 | 1.6 | 66.8 |
| Case 4 | −592.5 | −14.5 | −607.0 |

### 5.3.3. Cooperative Multi-objective Optimization Results

It can be concluded from Table 6 that, under the cooperative multi-objective optimization model, the total daily economic cost $C_{ETC}$ of MMGS is the lowest. The cost of case 4 adopting the cooperative multi-objective model is 16.3% lower than that for case 1, 13.9% lower than that for case 2, 8.6% lower than that for case 3 which only uses the economic dispatch model of the inner-loop without optimizing reactive power output of DC/AC converters. It is proved that the cooperative multi-objective optimization model improves the economy of MMGS.

**Table 6.** The final economic cost of MMGS in four cases.

| Case | Case 1 | Case 2 | Case 3 | Case 4 |
|---|---|---|---|---|
| $C_{ETC}$ Total Economic Cost of MMGS/CNY | 8585.7 | 8350.3 | 7862.7 | 7188.9 |

It can be concluded by analyzing the carbon emissions data in Table 7 that the total carbon emissions of the MMGS and distribution network with cooperative multi-objective optimization are the lowest among the four cases, which is 24.0% lower than that for case 1, 24.6% lower than that for case 2, and 29.8% lower than that for case 3, which does not optimize the reactive power. The economic cost of MMGS, the network loss of the distribution network, and the total carbon emission of MMGS and the distribution network were all optimized, which fully proves that the cooperative multi-objective optimization achieved the effect.

**Table 7.** Total carbon emissions in the four cases.

| Case | Case 1 | Case 2 | Case 3 | Case 4 |
|---|---|---|---|---|
| $E_C$ <br> Total Carbon Emissions/kg | 237.6 | 239.5 | 257.1 | 180.5 |

### 5.3.4. Further Verification

To further verify the correctness and validity of the model, the weather type of a certain day in the winter of December 2020 in Wuhan City, Hubei Province, China was used as the input of the model. The results of the economic dispatch of the inner-loop model are shown in Table 8, and the results of the network loss optimization of the outer-loop model are shown in Table 9. The network loss cost and energy loss cost in four cases are shown in Table 10. The final economic cost of MMGS in the four cases on another day is shown in Table 11.

**Table 8.** Comparison of operating cost in three cases on another day.

| Case | MGs | $C_{ex}$ <br> Energy Transaction Cost/CNY | $C_{ci}$ <br> Carbon Emissions Cost/CNY | $C_{OTC}$ <br> Total Operating Cost/CNY |
|---|---|---|---|---|
| | RMG | 2968.9 | 23.4 | 2992.3 |
| Case 1 | OBMG | 6446.4 | 48.6 | 6495.0 |
| | MMGS | 9415.3 | 72.0 | 9487.3 |
| | RMG | 2836.7 | 23.4 | 2860.1 |
| Case 2 | OBMG | 6304.5 | 44.6 | 6349.1 |
| | MMGS | 9141.2 | 68.0 | 9209.2 |
| | RMG | 2574.4 | 48.5 | 2622.9 |
| Case 3 | OBMG | 6154.2 | 27.6 | 6181.8 |
| | MMGS | 8728.6 | 76.1 | 8804.7 |

**Table 9.** Comparison of network loss in the four cases on another day.

| Case | Case 1 | Case 2 | Case 3 | Case 4 |
|---|---|---|---|---|
| $W_S^G$ <br> Total Network Loss/kW | 14,933.1 | 14,915.8 | 14,919.4 | 14,110.8 |
| $W_S^I$ <br> Increased Network Loss/kW | 145.2 | 127.9 | 131.5 | −677.1 |

**Table 10.** Network loss cost and energy loss cost on four cases on another day.

| Case | $C_{il}$ <br> Network Loss Cost/CNY | $C_{co}$ <br> Carbon Emissions Cost/CNY | $C_{WTC}$ <br> Energy Loss Cost/CNY |
|---|---|---|---|
| Case 1 | 107.4 | 2.6 | 110.0 |
| Case 2 | 94.6 | 2.3 | 96.9 |
| Case 3 | 97.3 | 2.4 | 99.7 |
| Case 4 | −501.1 | −12.3 | −513.4 |

**Table 11.** The final economic cost of MMGS in the four cases on another day.

| Case | Case 1 | Case 2 | Case 3 | Case 4 |
|---|---|---|---|---|
| $C_{ETC}$ <br> Total Economic Cost of MMGS/CNY | 9597.3 | 9306.1 | 8904.4 | 8291.3 |

It can be concluded from the above tables that the multi-objective optimization of the model is still achieved after using the weather data of one day in winter. The optimal total

economic cost of MMGS $C_{ETC}$ and the lowest distribution network loss $W_S^G$ are obtained, which further proves the correctness and effectiveness of the model.

## 6. Conclusions

A cooperative multi-objective optimization strategy for MMGS containing EVs and RESs is proposed, including dynamic economic dispatch and optimization of reactive power output by DC/AC converters. Dynamic economic dispatch optimizes the active power output of RESs, EVs, and DC/AC converters in MMGS to obtain the optimal daily operating cost of MMGS. Reactive power optimization reduces the daily operating network loss of the distribution network by optimizing the reactive power output of the DC/AC converters to the distribution network. By comparing the results of the four cases, the following conclusions are drawn:

1. According to the simulation results, the economic dispatch model of the inner-loop in the cooperative multi-objective optimization can reduce the operating cost of MMGS, which makes full use of the ATSET of EVs. Additionally, the optimization of the output reactive power output of the DC/AC converters of the outer-loop can reduce network loss cost and carbon emissions cost of the distribution network. The two cooperate to realize the improvement of the economy of MMGS and the efficient operation of the distribution network.
2. The cooperative multi-objective optimization model not only realizes the optimization of the economic cost of MMGS and the network loss of the distribution network, but also reduces the total carbon emissions of MMGS and the distribution network, which greatly responds to the calls for national carbon neutrality and carbon peak.

**Author Contributions:** Conceptualization, Z.X. and C.C.; methodology, Z.X.; software, Z.X.; validation, Z.X., M.D. and J.Z.; formal analysis, Z.X.; investigation, D.H.; resources, H.C.; data curation, Z.X.; writing—original draft preparation, Z.X.; writing—review and editing, Z.X. and C.C.; visualization, Z.X.; supervision, C.C.; project administration, C.C.; funding acquisition, C.C. All authors have read and agreed to the published version of the manuscript.

**Funding:** This research was funded by the National Natural Science Foundation of China, grant number 51977086.

**Institutional Review Board Statement:** Not applicable.

**Informed Consent Statement:** Not applicable.

**Data Availability Statement:** Not applicable.

**Conflicts of Interest:** The authors declare no conflict of interest.

## Nomenclature

| | |
|---|---|
| $B_{ij}$ | Susceptance of branch $ij$ in the distribution network. |
| $C_{ETC}$ | Economic total cost of MMGS. |
| $C_{OTC}$ | Operating total cost of MMGS from the inner-loop model. |
| $C_{WTC}$ | Energy loss cost of the MMGS from the outer-loop model. |
| $C_{il}$ | Network loss cost. |
| $C_{co}$ | Carbon emissions cost. |
| $C_{OCm}$ | Operating cost of the $m$-th MG from the inner-loop model. |
| $C_{exm}$ | Energy transaction cost in the $m$-th MG. |
| $C_{cim}$ | Carbon emissions cost in the $m$-th MG. |
| $C_{resm}$ | Cost of RESs of the $m$-th MG. |
| $C_{PVm}$ | Cost of PVs in the $m$-th MG. |
| $C_{WTm}$ | Cost of WTs in the $m$-th MG. |
| $C_{m,t}^{PV}$ | PV power generation cost in the $m$-th MG in the $t$-th hour. |
| $C_{m,t}^{WT}$ | WT power generation cost in the $m$-th MG in the $t$-th hour. |
| $C_{evm}$ | Cost of energy exchange between the $m$-th MG and EVs. |

| | |
|---|---|
| $C_{m,t}^{CEV}$ | Charging price of EVs in $m$-th MG in the $t$-th hour. |
| $C_{m,t}^{DEV}$ | Discharging price of EVs in $m$-th MG in the $t$-th hour. |
| $C_{gm}$ | Energy exchange cost between the $m$-th MG and the distribution network. |
| $C_{m,t}^{G}$ | Electricity price that $m$-th MG purchases/sells to the distribution network in the $t$-th hour. |
| $C_{cym}$ | Additional cycle cost of EV batteries in $m$-th MG. |
| $C_{cyn}^{EV}$ | Additional battery charging/discharging cycle cost of $n$-th EV. |
| $E_C$ | Total carbon emissions of MMGS and the distribution network. |
| $E_{Cim}$ | Carbon emissions generated by $m$-th MG in the inner-loop model. |
| $E_{CO}$ | Carbon emissions in the outer-loop model. |
| $e_c$ | Carbon emissions factor. |
| $f$ | The main objective function of the cooperative optimization model. |
| $f_1$ | The objective functions of the inner-loop model. |
| $f_2$ | The objective functions of the outer-loop model. |
| $G_{ij}$ | Conductance of branch $ij$ in the distribution network. |
| $I_{ij}^{max}$ | Upper limit of branch $ij$ current carrying capacity in the distribution network. |
| $i,j$ | Nodes of the distribution network. |
| $k_{il}$ | Loss cost coefficient. |
| $k_c$ | Carbon cost factor. |
| $k_{cy}$ | Number of additional charging/discharging cycles. |
| $k_i$ | The state variable of the $i$-th branch switch. |
| $M$ | Number of MGs in the MMGS. |
| $N$ | Number of EVs in $m$-th MG. |
| $N_{br}$ | Number of branches in the distribution network. |
| $P_{m,n,t}^{EV}$ | Exchanging power in the $t$-th hour of the $n$-th EV in the $m$-th MG. |
| $P_{m,t}^{EV}$ | Exchanging power in the $t$-th hour of the EVs in the $m$-th MG. |
| $P_{m,n,t}^{EVCDIs}$ | Power of the EVCDIs of the $n$-th EV in the $m$-th MG in the $t$-th hour. |
| $P_{m,t}^{PV}$ | Power output of PVs in the $m$-th MG in the $t$-th hour. |
| $P_{m,t}^{WT}$ | power output of WTs in the $m$-th MG in the $t$-th hour. |
| $P_{m,t}^{G}$ | Active power output between the $m$-th MG and the distribution network in the $t$-th hour through the DC/AC converters. |
| $P_{m,n,R}^{EVCDIs}$ | Rated power of the EVCDI serving the $n$-th EV in the $m$-th MG. |
| $P_{m,t}^{L}$ | Total load of the $m$-th MG in the $t$-th hour. |
| $P_{ij,t}$ | Active power of branch $ij$ in the $t$-th hour. |
| $P_{ij,t}^{0}$ | Initially active power of branch $ij$ when connected without MG in the $t$-th hour. |
| $P_{i,t}^{0}$ | Initial input active power of node $i$ in the $t$-th hour. |
| $P_{Li,t}$ | Active load of node $i$ in the $t$-th hour. |
| $P_{ij,max}$ | Maximum active power of the branch $ij$. |
| $Q_{ij,t}$ | Reactive power of branch $ij$ in the $t$-th hour. |
| $Q_{ij,t}^{0}$ | Initially reactive power of branch $ij$ when connected without MMGS in the $t$-th hour. |
| $Q_{m,t}^{G}$ | Reactive power output between the $m$-th MG and the distribution network in the $t$-th hour through the DC/AC converters. |
| $Q_{i,t}^{0}$ | Initial input reactive power of node $i$ in the $t$-th hour. |
| $Q_{Li,t}$ | Reactive load of node $i$ in the $t$-th hour. |
| $Q_{ij,max}$ | Maximum reactive power of the branch $ij$. |
| $R_{ij}$ | The resistance of branch $ij$. |
| $SOC_{EVm,n,t}$ | Remaining power capacity of the $n$-th EV in the $m$-th MG in the $t$-th hour. |
| $SOC_{EVm,n,min}$ | Minima capacity, respectively, of the $n$-th EV in the $m$-th MG. |
| $SOC_{EVm,n,max}$ | Maximum capacity, respectively, of the $n$-th EV in the $m$-th MG. |
| $S_m$ | Rated power of the DC/AC converter in the $m$-th MG. |
| $T$ | Scheduling cycle, one day, 24 h. |
| $V_{ij,t}$ | Voltage of branch $ij$ in the $t$-th hour. |
| $V_i^{min}$ | Lower limits of the node $i$ voltage amplitude. |
| $V_i^{max}$ | Upper limits of the node $i$ voltage amplitude. |
| $W_S^{B}$ | Original baseline network loss. |
| $W_S^{G}$ | Total daily operating network loss of the distribution network. |
| $W_S^{I}$ | Daily operating increased network loss of the distribution network. |



| | |
|---|---|
| σ | Self-discharge coefficient of EV's battery. |
| $\Delta t$ | Length of the time slot set for the optimization. |
| $\eta_{DEV}$ | Efficiency for EV discharging. |
| $\eta_{CEV}$ | Efficiency for EV charging. |
| $\delta_{ij}$ | Phase angle difference of branch $ij$. |

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
