# Peer review of "Cooperative Multi-Objective Optimization of DC Multi-Microgrid Systems in Distribution Networks"

_applsci, doi:10.3390/app11198916_

Round 1
Reviewer 1 Report
The paper deals with the implementation and results from an optimization model which optimizes the charge/discharge of EVs which are connected to two types of microgrids - a residential based micrgrid and an office park based microgrid. The optimization targets the minimization of operating costs when the microgrids are used in a multi-microgrid system. The paper is generally well written. The issue of organising the system as multi-microgrids is a topical one and therefore the subject of the paper is of interest.
The novelty lies mostly in the proposed use of two optimization loops - an inner which which does the economic dispatch and an outer one which optimizes reactive power flow to minimize losses. Individually the optimizations are relatively standard for this type of study.
Some references should be made to multi-micorgrid systems which is a more usual term in the literature compared to the term "Regional Joint Microgrid Systsm" as used by the authors. I would expect that the introduction gives some background and references to work done on multi-microgrid systems.
The abstract (and perhaps the title) should make clear that the work deals with DC microgrids.
Page 12 states that "The additional battery charging/discharging cycle cost of EV is 50 ¥ each time." - not clear what this means - dies this imply that there is a cost allocated to extra charge/discharge cycles? If so what is this based on?
In a related comment Case 3, although it has lowest cost seems to involve much deeper discharge of the EV batteries which would somehow represent an extra cost to the users. Please clarify if this cost is included - is this the additional charge/dischare cost referred to? For example Table 3 shows the cost of EV users - what exactly is included in this?
Colour of line in legend of fig. 7 (a) seems incorrect
My main concern is regarding the simplicity of the case study. The case study consists only of 2 microgrids - one office based and one residential based, although this may be OK for illustration pruposes.
However only one day of data is used for the study which makes it difficult to draw general conclusions. The use of a full year of data incorporating seasonal variation in wind and PV etc would be more typical for this kind of study. The single day results are useful for understanding what is happening, but general conclusions would need results for a year. The authors should augment the results with results for a year or at least with further results for other representative 24 hour periods.
Author Response
Comments and Suggestions for Authors
The paper deals with the implementation and results from an optimization model which optimizes the charge/discharge of EVs which are connected to two types of microgrids - a residential based micrgrid and an office park based microgrid. The optimization targets the minimization of operating costs when the microgrids are used in a multi-microgrid system. The paper is generally well written. The issue of organising the system as multi-microgrids is a topical one and therefore the subject of the paper is of interest.
The novelty lies mostly in the proposed use of two optimization loops - an inner which which does the economic dispatch and an outer one which optimizes reactive power flow to minimize losses. Individually the optimizations are relatively standard for this type of study.
Response:
The authors are grateful for your efforts made in reviewing our manuscript and understanding the significance of the topic addressed. All of your comments were taken into account and corresponding corrections were made. The paper has been improved and some details have been clarified.
(1) Some references should be made to multi-micorgrid systems which is a more usual term in the literature compared to the term "Regional Joint Microgrid Systsm" as used by the authors. I would expect that the introduction gives some background and references to work done on multi-microgrid systems.
Response:
Thank you for your valuable suggestion. The revised paper replaces the regional joint microgrid system (RJMS) with a DC multi-microgrid system (MMGS). In the introduction, references to the multi-microgrid system are introduced to illustrate some current researches on the multi-microgrid system. And in the second section: system structure, a detailed introduction to the bidirectional DC/AC converter in the DC microgrid is added to make the structure description of the DC microgrid more detailed. Both the abstract and the title have been changed to make the work of this paper more closely related to the DC microgrid.
(2) Page 12 states that "The additional battery charging/discharging cycle cost of EV is 50 ¥ each time." - not clear what this means - dies this imply that there is a cost allocated to extra charge/discharge cycles? If so what is this based on?
Response:
Thank you for the reviewer’s question on the additional battery charging/discharging cycle cost of EV. The battery of an EV has a service life, that is, the battery has a limit on the number of charging/discharging cycles. The scheduling cycle of this paper is one day. To simplify the analysis, this paper assumes that the EV will be fully charged at the end of the day. EV is discharged and then charged within a day, which is regarded as a normal battery charging/discharging cycle. In addition to the normal charging/discharging cycle, if the EV is discharged again, it will increase the number of additional cycles of battery charging/discharging. The additional charging/discharging cycles of the battery have an impact on the life of the battery and will generate additional battery charge/discharge cycle cost. MMGS needs to pay the corresponding cost to users of EVs for the additional battery charging/discharging cycle cost, and the cost incurred is called the additional battery charging/discharging cycle cost. The EV selected in this paper is BYD E6. Refer to the following research, the additional battery charging/discharging cycle cost is 50¥ per cycle.
- Bocca, A.; Baek, D. Optimal Life-Cycle Costs of Batteries for Different Electric Cars. In Proceedings of the 2020 AEIT International Conference of Electrical and Electronic Technologies for Automotive (AEIT AUTOMOTIVE), Turin, Italy, 18-20 November 2020; pp. 1-6.
(3) In a related comment Case 3, although it has lowest cost seems to involve much deeper discharge of the EV batteries which would somehow represent an extra cost to the users. Please clarify if this cost is included - is this the additional charge/discharge cost referred to? For example Table 3 shows the cost of EV users - what exactly is included in this?
Response:
To simplify the analysis, this paper does not consider the depth of discharge, only the number of cycles of the battery. The user cost of EVs mainly includes the cost of charging and discharging. EV charging will bring charging cost to users of EVs, while EVs discharge will reduce the charging cost of EVs. According to the trading principle, the discharging price of EVs should be greater than the charging price. EVs are discharged to MMGS, and users of EVs can make profits and reduce the cost of charging. The additional battery charging/discharging cycle cost is due to the scheduling of MMGS, which may cause the battery to produce additional battery charging/discharging cycles in addition to the normal charging/discharging cycle once a day. This cost is paid by MMGS to users of EVs, so it is ultimately the cost of MMGS, and users will not incur the cost. The charging cost of users is shown in Table 3. According to the scheduling strategy, EVs in MMGS can not generate additional charging/discharging cycles. If the cost is negative, it means that users have not paid the charging cost, but have made profits. Many references are based on this consideration:
- Chen, C.; Chen, J.; Wang, Y.; Duan, S.; Cai, T.; Jia, S. A price optimization method for microgrid economic operation considering across-time-and-space energy transmission of electric vehicles. IEEE Transactions on Industrial Informatics 2020, 16, 1873-1884.
- Leou, R.-C. Optimal charging/discharging control for electric vehicles considering power system constraints and operation costs. IEEE transactions on power systems 2015, 31, 1854-1860.
(4) Colour of line in legend of fig. 7 (a) seems incorrect
Response:
Thank you for the constructive suggestion and careful review. The colour of the line in the legend of fig. 7 (a) is revised.
(5) The case study consists only of 2 microgrids - one office based and one residential based, although this may be OK for illustration pruposes.
The main purpose of this paper is to explain the principle and prove the validity and correctness of the multi-objective cooperative optimization model. Therefore, two representative microgrids, RMG and OBMG, are selected to verify the feasibility of the method and model. According to the modeling method and optimization process, the method in this paper is also suitable for multi-microgrid systems.
(6) The single day results are useful for understanding what is happening, but general conclusions would need results for a year. The authors should augment the results with results for a year or at least with further results for other representative 24 hour periods.
Response:
Thank you for the constructive suggestion. The weather data of one day in the summer of June 2020 in Wuhan City, Hubei Province, China is used in the original paper to predict MMGS 24-hour renewable power generation. According to the reviewer’s suggestion, another representative 24-hour-a-day optimization results are added to the revised paper to prove the universal applicability of the optimization method and model. The added content is as follows:
5.3.4. Further Verification
To further verify the correctness and validity of the model, the weather type of a certain day in the winter of December 2020 in Wuhan City, Hubei Province, China is used as the input of the model. The results of the economic dispatch of the inner-loop model are shown in Table 8, and the results of the network loss optimization of the outer-loop model are shown in Table 9. The final economic cost of MMGS in four cases in another day is shown in Table 11.
Table 8. Comparison of operating cost in three cases in another day
|
Case |
MGs |
Cex Energy Transaction Cost/¥ |
Cci Carbon Emissions Cost/¥ |
COTC Total Operating Cost /¥ |
|
Case 1 |
RMG |
2968.9 |
23.4 |
2992.3 |
|
OBMG |
6446.4 |
48.6 |
6495.0 |
|
|
MMGS |
9415.3 |
72.0 |
9487.3 |
|
|
Case 2 |
RMG |
2836.7 |
23.4 |
2860.1 |
|
OBMG |
6304.5 |
44.6 |
6349.1 |
|
|
MMGS |
9141.2 |
68.0 |
9209.2 |
|
|
Case 3 |
RMG |
2574.4 |
48.5 |
2622.9 |
|
OBMG |
6154.2 |
27.6 |
6181.8 |
|
|
MMGS |
8728.6 |
76.1 |
8804.7 |
Table 9. Comparison of network loss in four cases in another day
|
Case |
Case 1 |
Case 2 |
Case 3 |
Case 4 |
|
Total Network Loss/kW |
14933.1 |
14915.8 |
14919.4 |
14110.8 |
|
Increased Network Loss/kW |
145.2 |
127.9 |
131.5 |
-677.1 |
Table 10. Network Loss Cost and Energy loss cost in four cases in another day
|
Case |
Cil Network Loss Cost/¥ |
Cco Carbon Emissions Cost/¥ |
CWTC Energy Loss Cost/¥ |
|
Case 1 |
107.4 |
2.6 |
110.0 |
|
Case 2 |
94.6 |
2.3 |
96.9 |
|
Case 3 |
97.3 |
2.4 |
99.7 |
|
Case 4 |
-501.1 |
-12.3 |
-513.4 |
Table 11. The final economic cost of MMGS in four cases in another day
|
Case |
Case 1 |
Case 2 |
Case 3 |
Case 4 |
|
CETC Total Economic Cost of MMGS/¥ |
9597.3 |
9306.1 |
8904.4 |
8291.3 |
It can be concluded from the above tables that the multi-objective optimization of the model is still achieved after using the weather data of one day in winter. The optimal total economic cost of MMGS CETC and the lowest distribution network loss are obtained, which further proves the correctness and effectiveness of the model.

Reviewer 2 Report
In this paper, the authors consider the management of microgrids with energy transmission of electrical vehicles (EV). They propose a model for optimizing a Regional Joint Microgrid System via Particle Swarm Optimization. They validate the approach testing a revised IEEE 33-node system.
Overall, the paper is well written, although some parts could be improved. While each individual idea (double-loop optimization, Particle Swarm Optimization…) is not new, the authors nicely combine these known ideas to implement valid algorithms for optimizing the management of a microgrid with EVs.
Major comments:
- Sometimes the manuscript is too schematic and its paragraphs too disconnected from the rest of the manuscript (e.g. 2.2).
- EVs are considered as pure storage systems. In minimising the costs, no user choices or preferences for instance about the time are considered. The resulting solution, although numerically optimal, may be unfeasible for users according to their needs.
- The formulation of the equations is strange and unreadable. There are too many superfluous variables and several variables have too many different names. There should only be one equality in each equation. In addition, there is little consistency between quantities such as vectors, single elements, and element wise scalar products, etc.
- The problem seems not multi-objective as it is perfectly decomposable in inner-loop and outer-loop. If this is not the case, the decomposition of the problem is not clear, and neither what weight one portion of the objective function has over the other.
Minor comments:
1) Page 5, line 37 it should be “which” instead of “that”.
2) Page 6, in the initialization there are two “din”. One should be “dout”.
3) Page 7, line 20, there are white spaces before the commas.
4) “kil” is defined in 4.6.1, however was already used in 4.6.
Author Response
Comments and Suggestions for Authors
In this paper, the authors consider the management of microgrids with energy transmission of electrical vehicles (EV). They propose a model for optimizing a Regional Joint Microgrid System via Particle Swarm Optimization. They validate the approach testing a revised IEEE 33-node system.
Overall, the paper is well written, although some parts could be improved. While each individual idea (double-loop optimization, Particle Swarm Optimization…) is not new, the authors nicely combine these known ideas to implement valid algorithms for optimizing the management of a microgrid with EVs.
Response:
The authors appreciate your time spent for reviewing this manuscript. Your comments have been carefully read and considered. The manuscript has been revised and improved according to your questions and suggestions.
(1) Sometimes the manuscript is too schematic and its paragraphs too disconnected from the rest of the manuscript (e.g. 2.2).
Response:
2.2 is a description of the topology of the DC microgrid. The newly added 2.3 is a detailed introduction to the bidirectional DC/AC converter in the DC microgrid. The DC/AC converter is a key part of the DC microgrid and can be used as a carrier for active and reactive power exchange between the DC microgrid and the distribution network. The contents of 2.2 and 2.3 are the basis for the construction of the multi-objective optimization model.
(2) EVs are considered as pure storage systems. In minimising the costs, no user choices or preferences for instance about the time are considered. The resulting solution, although numerically optimal, may be unfeasible for users according to their needs.
Response:
Thank you for the constructive suggestion and valuable question. As the reviewer noticed, the model proposed in this model includes some assumptions. To the view of the authors, the assumptions that occurred in this manuscript have been deliberated and the influence on the practicality of the method is limited.
In this paper, the commuting time of users is assumed to be fixed. The paper refers to related literature, as 77.95% of EVs' users will reach the working area at 7:30–9:30. And according to user habits, 90% of users will not change commuting habits. If a small number of users have an emergency, some extra backup batteries can be configured to provide services for these users, so it is assumed that the user's habit does not affect the verification of the method in the paper.
(3) The formulation of the equations is strange and unreadable. There are too many superfluous variables and several variables have too many different names. There should only be one equality in each equation. In addition, there is little consistency between quantities such as vectors, single elements, and element wise scalar products, etc.
Response:
Thank you for the constructive suggestions. The author has fully checked and revised the formulas and variables in the paper. In the revised paper, variable nomenclature is added before the introduction to visually display each variable. All the variables in the paper are reflected in the modeling and algorithm operation, and the redundant variables have been eliminated. In the revised paper, each equation contains only one formula. The writing of variables in the paper is uniformly defined, take "PEVm,n,t" as an example:
P stands for power, EV stands for the electric vehicle, m stands for the m-th microgrid in MMGS, n stands for the n-th electric vehicle in the m-th microgrid, and t stands for time, which is the t-th hour of the day. All variables are defined in this way, and there are no redundant variables.
(4) The problem seems not multi-objective as it is perfectly decomposable in inner-loop and outer-loop. If this is not the case, the decomposition of the problem is not clear, and neither what weight one portion of the objective function has over the other.
Response:
Thank you for your question. The model has two goals. The first is the lowest total economic cost CETC of MMGS, and the second is to achieve the lowest network loss of the distribution network . The optimization results of the inner-loop and outer-loop model are combined to achieve these two goals. The first is the total economic cost. The cost of the optimization of the inner-loop model and the cost of the optimization of the outer-loop model are combined to obtain the optimal total economic cost. However, the optimization conditions of the outer-loop model are derived from the inner-loop model. The inner-loop model optimizes the output of EV, PV, WT, and DC/AC converters at every moment. Among them, the active output of the DC/AC converter determines the optimization range of the outer-loop model. The outer-loop model optimizes the network loss of the distribution network by optimizing the reactive output of the DC/AC converters, which is the second goal of the model. Therefore, the inner-loop model and the outer-loop model cannot be completely separated. There is a connection between the two and they are combined.
Minor comments:
1) Page 5, line 37 it should be “which” instead of “that”.
2) Page 6, in the initialization there are two “din”. One should be “dout”.
3) Page 7, line 20, there are white spaces before the commas.
4) “kil” is defined in 4.6.1, however was already used in 4.6.
Response:
Actually, as the reviewer noticed, there are several format errors in the manuscript, including the misuse of punctuations and fonts. The authors have checked the whole passage and revised these errors.
- “that” is replaced with “which”
- Two "din" has been modified to "din" and "dout"
- The space before the comma is deleted.
- All variables are defined before the introduction.
